# Target Detection for Synthetic Aperture Radiometer Based on Satellite Formation Flight

**DOI:** 10.3390/s23146348

**Published:** 2023-07-12

**Authors:** Rui Li, Li Deng, Yuan Wang, Haoming Dai, Ran Duan

**Affiliations:** 1National Space Science Center, Chinese Academy of Sciences, Beijing 100190, China; 2School of Astronomy and Space Science, University of Chinese Academy of Sciences, Beijing 100049, China; 3School of Astronautics, Beihang University, Beijing 100191, China

**Keywords:** satellite formation, SAIR, spatial resolution, reconstructed image, target detection

## Abstract

Synthetic aperture interferometers formed by satellite formations have been adopted to improve spatial resolution. Due to the limited number of satellites and limited integrated time, the use of sparse baselines can result in distorted reconstructed images, which will generate false targets or miss true targets. When detecting a target on the Earth from a geostationary orbit, the target usually occupies only one pixel, and it is almost submerged by noise. Considering the slow-varying characteristics of the observation area, combined with historical observation data and the motion characteristics of the target itself, a target detection method based on multi-frame snapshot images is proposed. Firstly, the observation background is estimated using multi-frame historical data, and background elimination is used to suppress the background noise. Then, potential targets are selected using the local brightness temperature characteristics of the targets. Lastly, the target motion tracks are applied to erase false targets and correct the positions of missed targets. Simulation experiments have been conducted, and the false alarm rate and the missing alarm rate are counted for randomly distributed targets.

## 1. Introduction

A synthetic aperture interferometer radiometer (SAIR) is a typical imaging system that samples in a spatial frequency domain and obtains spatial domain images via inverse Fourier transform. SAIRs have the advantage of having fewer requirements in terms of weather conditions and operating time, making them widely used in meteorology, ocean exploration, and other remote sensing fields. The first on-board SAIR, the Microwave Imaging Radiometer with Aperture Synthesis (MIRAS) of the Soil Moisture and Ocean Salinity (SMOS) mission, worked on a low Earth orbit (LEO) of 600–800 km, covering a swath width of 1000 km, and had a ground spatial resolution of less than 50 km [1,2]. In order to provide continuous dynamic monitoring for regional areas, some geostationary orbit (GEO) SAIRs were proposed. These included GeoSTAR from the Jet Propulsion Laboratory of NASA [3], the Geostationary Atmospheric Sounder (GAS) from the European Space Research and Technology Center of the ESA [4], and the Geostationary Interferometric Microwave Sounder (GIMS) from the National Space Science Center, Chinese Academy of Sciences [5].

In recent years, SAIRs have started being used for target detection and recognition. Liu [6] demonstrated that the root-mean-square error of reconstructed images with targets with rapid brightness temperature (BT) changes, such as tropical cyclones (TCs), was closely related to observation frequency and imaging period. Chen [7] proposed a method for detecting higher-order moving targets using a rotating scanning SAIR (RS-SAIR) equipped with a linear sparse array. Yang and Hu [8] utilized the kernel method to predict the observation background by designing a new robust loss function, and a constant false alarm rate (CFAR) detector was used to detect targets. Lu [9] carried out extensive research on ship detection using an airborne SAIR, and airborne experiments were performed as well. Gao [10] proposed a small target detection method using a feature mapping neural network. The angular resolution of the on-board SAIR was determined by the diameter of the radiometer’s antenna. With the same angular resolution, the higher the orbit height is, the lower the ground resolution is. Especially in a high Earth orbit (HEO), the ground resolution can be several kilometers. Since targets with a size of hundreds of meters, such as cargo ships and oil tanks, usually occupy sub-pixels, the traditional target detection algorithms mentioned above are typically not applicable. 

To improve the angular resolution, the concept of satellite formation was introduced. AKS et al. [11] proposed an interferometric microwave radiometer concept using a satellite flight formation. By equipping each satellite with an SAIR, an unconstrained-aperture-size microwave radiometer could be synthesized in space. Mark et al. [12] explored the orbit mechanics and imaging performance for the definition and development of an Earth-observing satellite-swarm-based MISAR mission scenario. Apart from theoretical considerations, the Sun Radio Interferometer Space Experiment (SunRISE) by NASA is a mission using satellite formation to synthesize a large aperture radiometer [13,14]. The mission is scheduled to be launched no earlier than 2024, and an array of six toaster-size CubeSats, flying in a super-synchronous geosynchronous Earth orbit within 10 km of each other, will work together to study solar activity. In addition, the Discovering Sky at the Longest wavelength (DSL) project [15], conducted by the National Space Science Center, Chinese Academy of Science, is another mission. The DSL will consist of a mother satellite and 6–9 daughter satellites, flying on the same circular orbit around the Moon, and forming a linear interferometer array. 

Although satellite formation can improve the SAIR’s spatial resolution, the sparse sampling limited by the number of satellites becomes a new problem. For the SAIR, various antenna array configurations have different sampling features in the UV domain. The common configurations include a Y-shaped configuration with high sampling efficiency [16] and a circle-shaped configuration with high radiometric sensitivity [17]. However, when the spatial frequency domain is under-sampled, the reconstructed image will be aliased due to the high side-lobe of the point spread function (PSF). There are two typical effects. Firstly, when the samples of high-frequency components are not enough, the image will be blurred, and details will be missed. Secondly, when the samples of low-frequency components are not enough, the image will be sharpened. In this case, detecting and recognizing the targets, especially for one-pixel or even sub-pixel targets, becomes challenging from one snapshot.

In this paper, one method based on the reconstructed sequence of images, was put forward to detect and recognize the targets in the observed area, especially moving ships on the ocean. Using this method, the background of the observed area was estimated using multi-frame historical data, and we obtained the background eliminated images. Then, the targets were detected by utilizing the local extremum operation and motion tracks. 

The rest of this paper is organized as follows. In Section 2, the configuration design model for the satellite formation is described. Then, the target detection method is proposed in Section 3. The simulation experiment is conducted in Section 4, and the performance of the method is analyzed in Section 5. Lastly, Section 6 states the conclusions.

## 2. Synthetic Aperture Radiometer Based on Satellite Flight Formation

### 2.1. Configuration Design for Satellite Formation

An SAIR based on satellite formation is introduced to synthesize a large-aperture SAIR and improve the spatial resolution. The configuration of the satellite formation will affect the frequency domain sampling, which directly impacts the quality of the reconstructed image. Due to the limitations of satellites, the sampling is sparse. The (*u*, *v*) coverage percent of the frequency domain is typically selected as an optimal indicator. Next, the relative orbit elements are employed to construct the dynamic model of the satellite formation [18] and to compute the baseline vectors between the satellites. Then, the (*u*, *v*) coverage percentage based on relative orbit elements is established.

To detect moving ships on the ocean, the satellites are chosen to operate in the geostationary orbit for the continuous exploration of the observation area. For the convenience of inter-satellite communication and relative measurement, the configuration of a subsatellite circle-shaped distribution is selected. At the same time, the satellites of this configuration can provide good instantaneous (*u*, *v*) coverage. 

Assuming that the mother satellite has the orbit elements $\left(a_{0},e_{0},i_{0},\Omega_{0},\omega_{0},M_{0}\right)$ and the *k*th daughter satellite has the orbit elements $\left(a_{k},e_{k},i_{k},\Omega_{k},\omega_{k},M_{k}\right)$, the relative orbit elements can be calculated using Equation (1). $n_{*}=\sqrt{\mu /a_{*}^{3}}$ is the mean motion of the satellite; *D* denotes the relative average drift rate; $\Delta e=(\Delta e_{x},\Delta e_{y})$ represents the relative eccentricity vector; $\Delta i=(\Delta i_{x},\Delta i_{y})$ is the relative inclination vector; and ∏ is the difference in the mean argument of latitude.
(1)$$\left\{\begin{array}{cc} D & =n_{k}-n_{0}\\ \Delta e_{x} & =e_{k}\cos\omega_{k}-e_{0}\cos\omega_{0}-e_{k}\sin\omega_{k}\Delta \Omega \cos i_{0}\\ \Delta e_{y} & =e_{k}\sin\omega_{k}-e_{0}\sin\omega_{0}+e_{k}\cos\omega_{k}\Delta \Omega \cos i_{0}\\ \Delta i_{x} & =(\Omega_{k}-\Omega_{0})\sin i_{0}\\ \Delta i_{y} & =-(i_{k}-i_{0})\\ \prod & =(\omega_{k}-\omega_{0})+(M_{k}-M_{0})+(\Omega_{k}-\Omega_{0})\cos i_{k} \end{array}\right.$$

As Figure 1 shows, around the mother satellite, the subpoints of the daughter satellites are distributed in a circle with a radius of r, and the initial phase of the *k*th satellite is $\phi_{k}$. The subsatellite plane track of the *k*th satellite is shown in Equation (2):(2)$$\left\{\begin{array}{l} x_{k}=r\cos\phi_{k}\\ y_{k}=r\sin\phi_{k} \end{array}\right.\;$$

Equation (2) is equivalent to the relative orbit elements $\left(D_{k},\Delta e_{xk},\Delta e_{yk},\Delta i_{xk},\Delta i_{yk},\prod_{k}\right)$. The formation’s stability requires $D_{k}=0$ and $\prod_{k}=0$, and the remaining elements are listed as follows:(3)$$\left\{\begin{array}{l} \Delta e_{xk}=-\frac{r}{2a}\sin\phi_{k}\\ \Delta e_{yk}=-\frac{r}{2a}\cos\phi_{k}\\ \Delta i_{xk}=\frac{r}{a}\sin\phi_{k}\\ \Delta i_{yk}=\frac{r}{a}\cos\phi_{k} \end{array}\right.$$

Therefore, by configuring the different $\phi_{k}$ of the satellite, we can obtain different samples in the (*u*, *v*) domain, as presented in Equation (4): (4)$$\;\left\{\begin{array}{l} u_{ij}=(x_{i}-x_{j})/\lambda \\ v_{ij}=(y_{i}-y_{j})/\lambda \end{array}\right.\;,\;i,j=1,2,\ldots ,N_{s}\;$$

Building on the optimization objective function proposed by Cornwell [19], a new optimization function is introduced in Equation (5):(5)$$m(r_{1},r_{2},\ldots ,r_{n})=\sum_{i,j,k,l}\log(\left|\sqrt[{3}]{u_{i,j}}-\sqrt[{3}]{u_{k,l}}\right|)$$

From Equation (5), we can obtain the formation configuration solution with the optimal (*u*, *v*) coverage.

### 2.2. Resolution and Sensitivity with Imaging

In this section, we firstly discuss the principle of SAIR imaging, and then the resolution and sensitivity of the SAIR are introduced.

#### 2.2.1. Imaging

The imaging principle of SAIR is based on the van Cittert–Zernike theorem. The theorem essentially states that the spatial coherence function, also called visibility, is exactly proportional to the Fourier components of the brightness. The visibility can be measured via cross-correlation between a pair of spatially separated antenna elements. The spatial domain is represented by the direction cosines of the incidence angle (ξ,η)=(sinθcosφ,sinθsinφ), and the spatial frequency domain is represented by baselines (u,v), which represent the distance between antennas measured in wavelength. The visibility is defined as follows:(6)$$V\left(u,v\right)=K\iint_{\xi^{2}+\eta^{2}\leq 1}G(\xi ,\eta )\frac{T_{B}(\xi ,\eta )}{\sqrt{1-\xi^{2}-\eta^{2}}}{\tilde{r}}_{h}(-\frac{u\xi +v\eta }{f_{0}})e^{-j2\pi (u\xi +v\eta )}d\xi d\eta$$
where *K* is a constant related to the receiver’s characteristics and the system bandwidth. G(ξ,η) is the antenna power pattern. $T_{B}(\xi ,\eta )$ is the brightness temperature in units of Kelvin. ${\tilde{r}}_{h}$ is the fringe washing function, and $f_{0}$ is the central frequency. 

By defining for simplicity,
(7)$$T(\xi ,\eta )=G(\xi ,\eta )\frac{T_{B}(\xi ,\eta )}{\sqrt{1-\xi^{2}-\eta^{2}}}$$
and neglecting spatial de-correlation effects (i.e.,${\tilde{r}}_{h}=1$), the image reconstruction of the brightness temperature map can be obtained by computing the inverse Fourier transform of the measured visibility:(8)$$T(\xi ,\eta )=\frac{1}{K}\int_{-\infty }^{+\infty }\int_{-\infty }^{+\infty }V(u,v)e^{j2\pi (u\xi +v\eta )}dudv$$

However, with a limited number of antennas, the (*u*, *v*) plane is only sampled at discrete points, resulting in severe degradation in the reconstructed image due to the inadequate frequency space sampling.

#### 2.2.2. Spatial Resolution and Sensitivity

The satellite formation with SAIR can form a spatial interferometric array, and the spatial resolution is determined by the formation distance:(9)$$res\propto \frac{\lambda }{D}$$

As Equation (9) shows, the spatial resolution is inversely proportional to the aperture size, where *res* is the spatial resolution, λ is the detection wavelength, and *D* is the diameter of the antenna opening. For a circle-shaped subsatellite distribution, *D* will be equal to 2*r* when two satellites are symmetrically distributed in the circle.

For a single radiometer, the sensitivity is as follows:(10)$$△T=\frac{T_{s}}{\sqrt{B\tau_{a}}}=\frac{T_{A}+T_{R}}{\sqrt{B\tau_{a}}}$$
where $T_{s}$ represents the system noise temperature, $T_{R}$ represents the receiver noise temperature, $T_{A}$ represents the average brightness temperature, *B* is the signal bandwidth of the receiver, and $\tau_{a}$ is the integrated time for the visibility data.

Considering SAIR formed by a satellite formation, if the SAIR has $n_{a}$ antennas, the visibility data are averaged for time $\tau_{a}$, and the whole observation covers a time interval $\tau_{0}$, the sensitivity of this array is as follows:(11)$$△T_{syn}=\frac{T_{A}+T_{R}}{\sqrt{n_{a}(n_{a}-1)B\tau_{0}}}\frac{A_{syn}}{A}$$
where *A* is the effective collecting area of the elemental antenna and $A_{syn}$ is the effective area of the beam created by the antenna array. By increasing the number of antennas $n_{a}$ and extending the time $\tau_{a}$, better sensitivity can be obtained. 

#### 2.2.3. Image SNR

The targets that are expected to be detected are moving ships. Due to the spatial resolution, each target only occupies one pixel or subpixel, and they are considered as point targets. The reconstrued image of sparse sampling has a worse signal-to-noise (*SNR*). Therefore, the neighborhood of the potential targets is defined, and the *SNR* within the neighborhood is calculated, instead of the *SNR* of the whole image. The *SNR* of a potential target’s neighborhood is calculated as follows:(12)$$SNR=10\mathrm{lg}(\frac{\sum \left|\left|T_{\mathrm{tar}}\right|-\left|T_{\mathrm{around}}\right|\right|}{\sum \left|T_{\mathrm{around}}\right|})$$
where $T_{\mathrm{tar}}$ is the brightness temperature of the potential target and $T_{\mathrm{around}}$ is the average brightness temperature of the potential target’s neighborhood, with a selection of an 11×11 pixel area.

## 3. Target Detection Algorithm and Process

The synthetic aperture radiometer based on satellite flight formation involves several key technologies, including satellite formation design and high-precision relative position measurement, the calibration of the instruments’ amplitude and phase to maintain consistency and stability, and image reconstruction and target detection on sparse sampling. Target detection, especially for point targets on sparse sampling, is one of the key technologies. This paper is focused on the target detection of moving ships using sparse sampling. 

All the frequency components were expected to be sampled to achieve a realistic reconstruction of the image, as Equation (8) indicates. Otherwise, the reconstructed image would be severely degraded, leading to various negative effects, such as generating false targets and missing true targets due to sparse baselines and target movement. These negative effects would make it difficult to detect targets from a single snapshot. However, if a series of snapshots of the observed area could be gained in a short period of time, the background remains relatively consistent across each snapshot, with the only difference being the position of the moving targets. 

Under these conditions, a target detection method based on the image sequences was put forward. Firstly, background estimation and elimination operations were performed, which could remove the aliasing noise caused by background elimination to some extent. Then, a local extremum operation was applied to detect potential targets. Lastly, targets were selected based on motion tracking to remove the noise caused by the targets’ movement. The overall process is illustrated in Figure 2, with varying colors indicating different levels of brightness temperature, and the targets to be detected are flagged with rectangular boxes.

### 3.1. Background Estimation and Elimination

Assuming that at moment k, targets have appeared in the observation area. In a short period of time, the background brightness temperature of the observation area ${\hat{T}}_{\mathrm{back}}$ can be estimated using the historical data and the average value of the *N*-frame reconstructed images during that period, as Equation (13) shows: (13)$${\hat{T}}_{\mathrm{back}}=\frac{1}{N}(\sum_{j=1}^{N}T^{j})$$

Taking into account the slow-varying characteristics of the observation area over a short and continuous period, the background elimination operation is utilized to remove the background interference. As Equation (14) shows, $T_{\mathrm{obj}}^{k}$ is the result after the background elimination operation:(14)$$T_{\mathrm{obj}}^{k}=T^{k}-{\hat{T}}_{\mathrm{back}}$$

### 3.2. Potential Target Selection Based on Local Extremum Operation

After performing the background elimination operation, noise caused by the movement of the targets appears, and the aliasing noise of the targets still exists. Considering the brightness temperature characteristics of the targets, a local extremum operation is used to select potential targets. The dilation operation is firstly applied to the image. We choose the structing element *B*, which focuses on the 7×7 neighborhood of the central element, and the elements in *B* are all ones, except for the center element, which is zero.
(15)$$B={\left(\begin{array}{ccccc} 1 & \cdots & & \cdots & 1\\ & 1 & 1 & 1 & \\ \cdots & 1 & 0 & 1 & \cdots \\ & 1 & 1 & 1 & \\ 1 & \cdots & & \cdots & 1 \end{array}\right)}_{7\times 7}$$
(16)$$T_{dilation}^{k}(l,m)=\underset{(l^{\prime },m^{\prime }):B(l^{\prime },m^{\prime })=1}{\max}\mathrm{sgn}(c)T_{\mathrm{obj}}^{k}(l+l^{\prime },m+m^{\prime })$$
(17)$$T_{extremum}^{k}(l,m)=\left\{\begin{array}{l} \left|T_{\mathrm{obj}}^{k}(l,m)\right|\;,\mathrm{sgn}(c)T_{\mathrm{obj}}^{k}(l,m)>T_{\mathrm{dilation}}^{k}(l,m)\\ 0\;\;\;\;\;\;\;\;\;\;\;\;\;\;\;\;,\mathrm{sgn}(c)T_{\mathrm{obj}}^{k}(l,m)\leq T_{\mathrm{dilation}}^{k}(l,m) \end{array}\right.$$
(18)$$\left\{P_{l,m}^{k}\right\}=\left\{\left(l,m,T_{\mathrm{extremum}}^{k}(l,m)\right)\right\}\;s.t.\;T_{\mathrm{extremum}}^{k}(l,m)>\vartheta \cdot \sigma (T_{\mathrm{extremum}}^{k})$$

Equations (16) to (18) depict the process of selecting potential targets, where $c\in \left\{-1,+1\right\}$, $T_{\mathrm{dilation}}^{k}(l,m)$ is the result after dilating, and $T_{\mathrm{extremum}}^{k}(l,m)$ is the non-extremum suppression result of the image. When the parameter *c* is set to −1, the local minimum points are selected; otherwise, the local maximum points are selected. $P_{l,m}^{k}$ records the position (l,m), and the local extremum value $T_{\mathrm{extremum}}^{k}(l,m)$ of the potential target. The number of potential targets $\left\{P_{l,m}^{k}\right\}$ is closely related to the detection threshold value ϑ and the standard deviation $\sigma (E_{\mathrm{obj}}^{k})$. If a low threshold value ϑ is chosen, all targets can be detected, but it will treat noise points as targets; otherwise, if a high threshold value is chosen, it may lead to the incomplete detection of targets. A reasonable threshold value needs to be set based on the target and noise characteristics.

It should be noted that the contrast in brightness temperature between targets and aliasing noise in certain frames may decrease after background elimination, which may lead to most targets being missed. To address this issue, we calculate the sum of the local extremum values $T_{sumextremum}^{k}$ of potential targets in each frame,
(19)$$T_{sumextremum}^{k}=\sum_{\left\{P_{l,m}^{k}\right\}}T_{extremum}^{k}(l,m)$$
and analyze the consecutive frames $\left\{T_{sumextremum}^{k}\right\}$: if outliers exist, the corresponding frame will be considered as a missed-detection moment, and all potential targets in this frame will be neglected.

### 3.3. Target Confirmation Based on the Motion Track

The motion characteristics of the potential targets are combined to confirm their validity. Herein, we introduce the hypothesis that the target’s motion track is continuous without any abrupt changes. Based on this hypothesis, a matching matrix $M_{k}^{k+1}$ is built for potential targets in two adjacent frames at non-missed-detection moments. The matrix has the same number of rows as the number of $\left\{P_{l,m}^{k}\right\}$ and the same number of columns as the number of $\left\{P_{l,m}^{k+1}\right\}$.
(20)$$M_{k}^{k+1}=\left(\begin{array}{cccc} a_{11} & \ldots & a_{1j} & \cdots \\ \vdots & \ddots & \vdots & \vdots \\ a_{i1} & \cdots & a_{ij} & \cdots \\ \cdots & \cdots & \cdots & \ddots \end{array}\right)$$
(21)$$a_{ij}=\left\{\begin{array}{l} \left|\left(l_{i}^{k}-l_{j}^{k+1},m_{i}^{k}-m_{j}^{k+1}\right)\right|e^{i\cdot \mathrm{atan}2(l_{i}^{k}-l_{j}^{k+1},m_{i}^{k}-m_{j}^{k+1})}\;\;\;\left|\left(l_{i}^{k}-l_{j}^{k+1},m_{i}^{k}-m_{j}^{k+1}\right)\right|<\upsilon_{\max}\cdot \tau \\ -1\;\;\;\;\;\;\;\;\;\;\;\;\;\;\;\;\;\;\;\;\;\;\;\;\;\;\;\;\;\;\;\;\;\;\;\;\;\;\;\;\;\;\;\;\;\;\;\;\;\;\;\;\;\;\;\;\;\;\;\;\;\;\;\;\;\;\;\;\;\left|\left(l_{i}^{k}-l_{j}^{k+1},m_{i}^{k}-m_{j}^{k+1}\right)\right|\geq \upsilon_{\max}\cdot \tau \end{array}\right.$$

The parameter $a_{ij}$ in the matrix records the relationship between the ith potential target in $\left\{P_{l,m}^{k}\right\}$ and the jth potential target in $\left\{P_{l,m}^{k+1}\right\}$. The calculation method of $a_{ij}$ is shown in Equation (21), where $\left(l_{i}^{k},m_{i}^{k}\right)$ is the position of the ith target and $\left(l_{j}^{k+1},m_{j}^{k+1}\right)$ is the position of the jth target; $\upsilon_{\max}$ is the maximum velocity of targets; τ is the time interval between two frames; and $a_{ij}$ can be seen as the velocity of the target. For a certain target, the velocity varies gradually, without any sudden changes.

The matching matrix $M_{k}^{k+1}$ records the motion information of potential targets. *K* + 1 frames of images will generate *K* matching matrices. By analyzing the matching matrix series $\left\{M_{k}^{k+1}\right\}$, the target’s motion track can be obtained. Furthermore, the target’s position can be amended, and the target’s position at missed-detection moments can be estimated using the motion track.

## 4. Simulation

### 4.1. Satellite Formation and Geometry Configuration

A SAIR composed of 11 satellites, which consists of 1 mother satellite and 10 daughter satellites, was used for the simulation. Each satellite carried a Y-shaped SAIR to explore the observation area. And the frequency was assigned to the K-band as an example. In order to satisfy the interferometric requirements and the baseline measurement precision, the time synchronization between satellites should be better than half of the wavelength. In addition, each satellite was equipped with a high-stability atomic clock and a GNSS receiver and was also equipped with a communication antenna to establish the inter-satellite link. All daughter satellites communicate with the mother satellite. Differential positioning technology based on GNSS was used to gain precious and accurate relative positioning amongst the satellites. All daughter satellites receive the calibration signal sent by the mother satellite to perform the amplification and phase calibration. The related engineering solutions refer to the research work in [15,20].

The flight formation of the satellites was maintained in a circular radius of approximately 1 km, and the phases of the daughter satellites could be acquired by optimizing Equation (5). The optimization result, as well as the orbit elements, are presented in Table 1 and Table 2, respectively. In each satellite, the antenna aperture of the Y-shaped SAIR was 0.15 m, with an arm length of 1.5 m. The equivalent synthetic aperture of the formation system could reach 2 km. The satellite formation in subsatellite view and the baselines are shown in Figure 3. The samples in the (*u*, *v*) domain in 20 s only covered 1.46% and 8.75% in 12 h. 

### 4.2. Simulation Scenario

Fifty moving ships with hundred-meter scales were randomly distributed on the ocean. The brightness temperature of the ocean was about 292~302 K, and the brightness temperature of the ships was about 250~260 K. The targets moved at a uniform linear speed of 20~25 knots during a short observation interval. The integrated time was 5 s, and the observation interval was about 20 s. That means the frame frequency of the imaging was 20 s. We used 12 h historical visibility sampling data without targets and 20 s visibility sampling data with targets to generate the reconstructed image.

The antenna aperture size was 0.15 m with a 4.77° field of view (FOV), enabling it to observe an area with a diameter of 3000 km on the Earth’s surface. Therefore, the spatial resolution of the system was 3.6 × 10^−4^°, and the ground resolution was 227 m. Table 3 listed the primary parameters of the simulation scenario.

We conducted multiple sets of random experiments, and each experiment generated 10 dynamic frames with targets moving 1 pixel per frame. The simulation period was about 200 s.

Taking one set of experiments for analysis, the brightness temperature of the observation area is shown in Figure 4a, and the reconstructed brightness temperature image using 12 h historical sampling data is shown in Figure 4b. 

In Figure 5, ten-frame original images are shown in column (a), and two of the targets in part of the observation area are flagged with rectangles as the examples. Column (b) depicts the images reconstructed by the SAIR based on the satellite flight formation. These two targets were buried in the noise. Column (c) shows the images after performing the background elimination operation, and potential targets are highlighted in the images. But they were not easy to be detected. Column (d) shows the potential targets after the local extremum operation. Finally, the motion characteristics of 10 images were combined, the targets were confirmed, and the trajectories were determined, as shown in Figure 6.

### 4.3. Target Detection Performance

A total of 30 sets of simulation experiment were conducted to test the method’s performance. Herein, despite the *SNR*, the false alarm rate and the missing alarm rate were introduced to analyze the detection results. The false alarm rate refers to the percentage of false targets detected among all targets detected, and the missing alarm rate is defined as the percentage of missed targets among all true targets. In Figure 7, the three lines are, respectively, the average *SNR* of the reconstructed images, the images after the background elimination operation, and the images after the local extremum operation. The blue curve represents the average *SNR* of the reconstructed images, the green curve represents the average *SNR* of the images after background elimination, and the red curve represents the average *SNR* of the images after obtaining the local extremum positions. The local extremum operation was one operation in the local area that remained at the minimum value and set the other values to zero. Using the local extremum operation, the noise was suppressed, and the *SNR* was improved.

The average false alarm rate and average missing alarm rate in 30 sets of experiments are shown in Figure 8.

## 5. Discussion

The target detection process included background elimination, a local extremum operation to detect potential targets, and an image series to confirm the targets.

The specific threshold during the local extremum operation was determined using the normalized reconstructive images. In this paper, we set 80 as a certain threshold ϑ. Comparatively, using a threshold of 70, the potential targets would be in the thousands. This was not consistent with the simulation scenario. Using a threshold of 90, the potential targets would be in the dozens, and it would lose the potential targets. So, 80 was selected for the simulation scenario.

As Figure 6 shows, the average *SNR* of the reconstructed image series was −47.8 dB. After background elimination, the average *SNR* turned out to be 9.0 dB. And after the filtering operation, the average *SNR* turned out to be 19.15 dB. The background elimination operation could significantly enhance the *SNR* of the targets in the degradation images, and the operation using local extremum knowledge could improve the *SNR* further. In the 5th, 16th, 17th, and 24th set of experiments, more targets were submerged by target movement noise and target brightness temperature difference noise in some frames after background elimination, and the missing alarm rate was above 10%, especially in the 22nd experiment, which had the worst missing alarm rate of about 24%. The method performed well in most experiments; the average missing alarm rate in all experiments was 5.2%, and the average false alarm rate was 4.8%.

## 6. Conclusions

A target detection method was proposed for a synthetic aperture radiometer based on satellite flight formation. Considering the slow-varying characteristics of the observation area and combined with historical observation data and the motion characteristics of the targets, the method was an effective means to solve the problem caused by sparse sampling. The errors such as baseline measurement errors and phase errors between satellites make the detection more difficult, and related research will be performed in the future. 

## Figures and Tables

**Figure 1 sensors-23-06348-f001:**
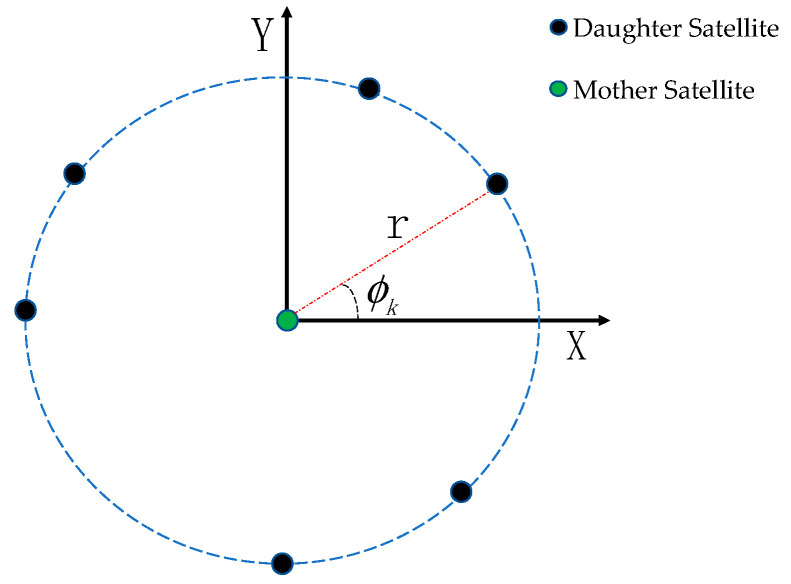
Schematic diagram of circle-shaped subsatellite distribution.

**Figure 2 sensors-23-06348-f002:**
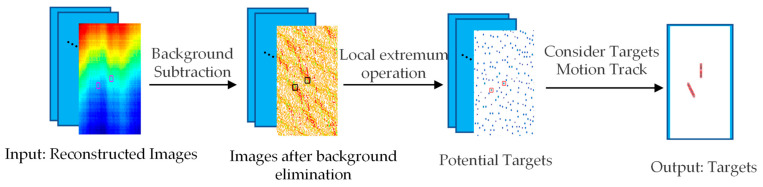
Detection process of the algorithm.

**Figure 3 sensors-23-06348-f003:**
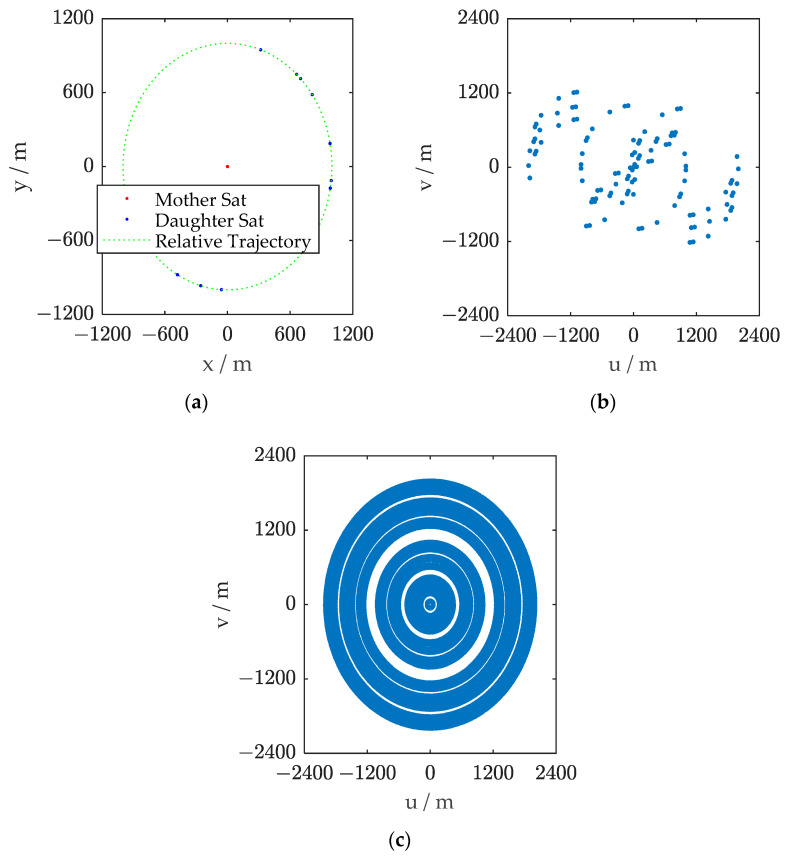
(**a**) The ground track of the formation. (**b**) The baseline distribution of the satellite formation in 20 s. (**c**) The baseline distribution of the satellite formation in 12 h.

**Figure 4 sensors-23-06348-f004:**
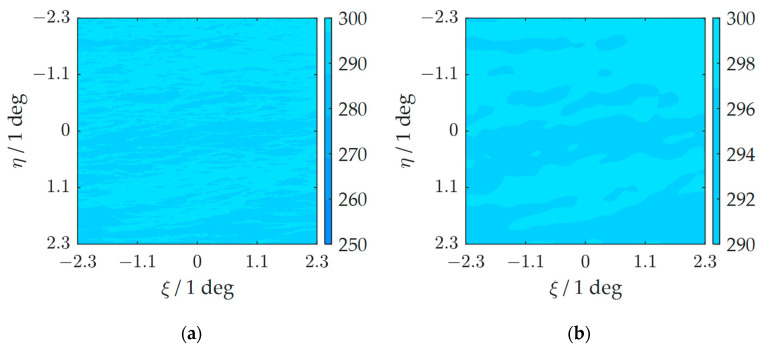
(**a**) When targets exist in the area, the brightness temperature distribution of the observation area. (**b**) The reconstructed image of the observation area.

**Figure 5 sensors-23-06348-f005:**
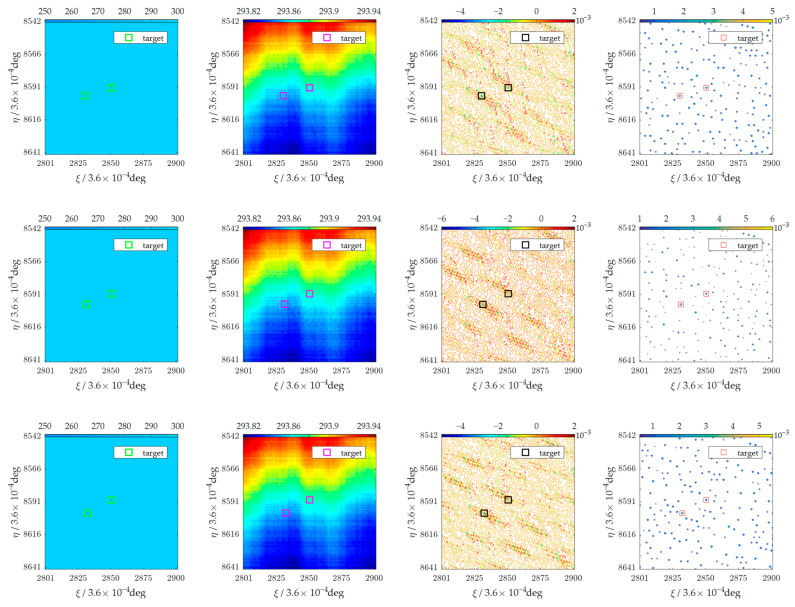
Ten frames of a set of experiments. (**a**) A region of the observation area with targets. (**b**) The reconstructed area. (**c**) The results of background elimination. (**d**) The results after the local extremum operation.

**Figure 6 sensors-23-06348-f006:**
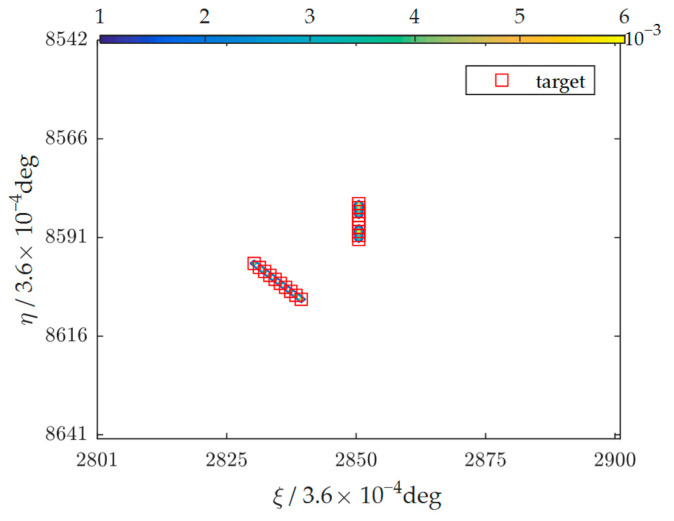
The confirmed targets and their tracks over ten frames.

**Figure 7 sensors-23-06348-f007:**
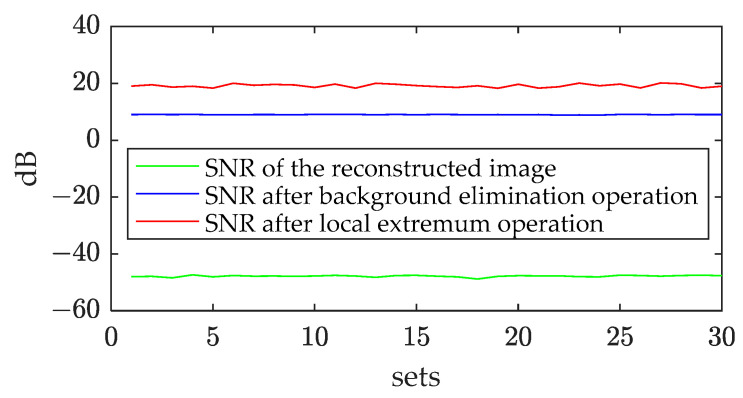
The average *SNR* of multiple sets of experiments.

**Figure 8 sensors-23-06348-f008:**
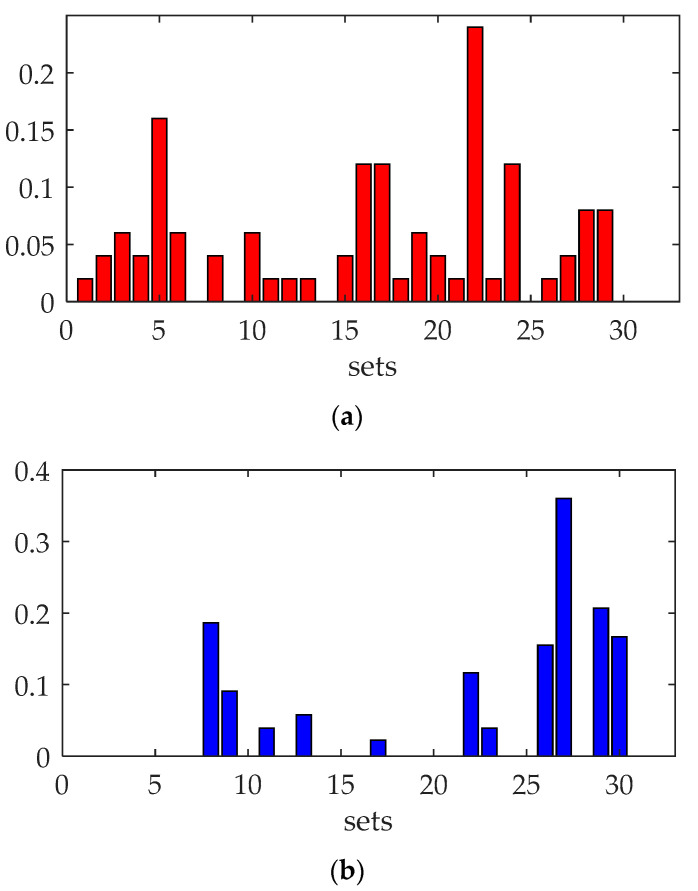
The detection results in multiple sets of experiments. (**a**) Missing alarm rate of the experiments. (**b**) False alarm rate of the experiments.

**Table 1 sensors-23-06348-t001:** Optimization results of the daughter satellites.

Parameter	Sat1	Sat2	Sat3	Sat4	Sat5	Sat6	Sat7	Sat8	Sat9	Sat10
ϕ **/deg**	10.82	35.78	45.59	48.52	74.51	241.37	255.15	266.68	349.93	353.47

**Table 2 sensors-23-06348-t002:** Orbital Elements of the daughter satellites.

Orbital Elements	a/km	e	i/deg	ω/deg	Ω/deg	M/deg
Sat0	42,164	0	0	0	0	0
Sat1	42,164	2.668 × 10^−6^	0.00136	90.000	−79.183	−10.82
Sat2	42,164	2.668 × 10^−6^	0.00136	90.000	−54.224	−35.78
Sat3	42,164	2.668 × 10^−6^	0.00136	90.000	−44.401	−45.59
Sat4	42,164	2.668 × 10^−6^	0.00136	90.000	−41.476	−48.52
Sat5	42,164	2.668 × 10^−6^	0.00136	90.000	−18.489	−74.51
Sat6	42,164	2.668 × 10^−6^	0.00136	90.000	151.370	−241.37
Sat7	42,164	2.668 × 10^−6^	0.00136	90.000	165.155	−255.15
Sat8	42,164	2.668 × 10^−6^	0.00136	90.000	176.683	−266.68
Sat9	42,164	2.668 × 10^−6^	0.00136	90.000	259.928	−349.93
Sat10	42,164	2.668 × 10^−6^	0.00136	90.000	263.486	−353.47

**Table 3 sensors-23-06348-t003:** Main parameters of the simulation scenario.

Parameter	Frequency	FOV	Antenna Size	Aperture Size of SAIR	Resolution
	K-band	4.77°	0.15 m	2 km	3.6 × 10^−4^°

## Data Availability

The data used to support the findings of this study are available from the corresponding author upon request.

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
