# Peer review of "Target Detection for Synthetic Aperture Radiometer Based on Satellite Formation Flight"

_sensors, 2023, doi:10.3390/s23146348_

Round 1

Reviewer 1 Report

Review of manuscript "Target Detection for Synthetic Aperture Radiometer Based on 2 Satellite Formation Flight" by Rui Li et. al.

The paper presents a method to detect small sources present within a large image retrieved by an interferometric radiometer. Due to imaging artifacts the said sources (or targets as the authors name it) are buried in noise. The method to highlight them consists of subtracting the background (supposedly known) using knowledge of previous images from historical data. The method apparently is intended to detect ships on the ocean, although this is not clearly stated, only guessed at the end of the paper. It would be nice to explain it at the beginning. Nevertheless, the simulation results included are not convincing (figure 6).

The measurement instrument is an interferometric radiometer made of several geostationary spacecrafts in flight formation, each one carrying an interferometric radiometer itself. This solution is aiming at achieving high spatial resolution, but the analysis does not mention important challenges of using satellite formation, as for example synchronization, syntonization or phase calibration among others. These effects introduce measurement errors that can be larger than the ones associated to uv coverage, which are the only ones considered in the paper.

In general the paper is difficult to understand and does not give a good impression to the reader regarding the feasibility of the proposal. 

Some comments follow

Section 2.1 is very obscure, at least for this reviewer. What the authors are trying to explain in it? T this reviewer it is just a collection of formulas without specific application in the rest of the paper. 

Equation (7) is wrong. There is no factor 2 in the denominator \Delta T= T_s/\sqrt{B \tau_a}

Equation (14) What is T_noise? Where this noise comes from?

Equation (15) What is the "target" and what the "around". In other words, how do you define a "target"?

Line 175: the sentence "From formular (14), we know that the true image can be reconstructed perfectly when all the frequency components are sampled" is not true. You would need an instrument with infinite size. 

Line 178: "... make the image large enough that there are no sources of interest...". What are the "sources of interest". To better understand the proposed method, the authors must explain the specific application of the instrument. 

Line 186: Why the target should move?

Line 269: "Each satellite carries a Y-shaped SAIR, ...". How do you synchronize and syntonize the baselines across satellites?. How do you keep the antenna separation constant in those baselines?. How you calibrate the phase of all baselines?. Without all these details, this instrument, based on satellite flight formation, will not work. 

Figure 3 (a): I guess the red point is the mother and the blue points the daughter (not the other way around)

Line 270: Again, what are the targets?? Are they ships? So you are proposing a system to detect ships on the ocean?. You should start by this and then deduce the instrument requirements to achieve this goal.

Figure 5. Apparently the background elimination does not clearly show the positions of the targets. They are really hard to see (if seen at all) in the rightmost column.

Figure 6. What do you mean by "local extremum operation"

Section 6 is more a summary of the paper than the statement of conclusions.

Author Response

Thanks for the valuable and constructive comments that are helpful for revising and improving our manuscript. Synthetic aperture radiometer based on satellite formation flight had several key technologies that include satellite formation design and high-precision relative position measurement, the instruments amplitude and phase calibration to keep consistency and stability, and image reconstruction and targets detection on sparse sampling. Targets detection especially for point targets on sparse sampling was one of the key technologies and was the research work in our paper. The paper focused on the moving ship targets detection with sparse sampling. The research background was added and explained on the introduction on the revised manuscript.

Comment 1: Section 2.1 is very obscure, at least for this reviewer. What the authors are trying to explain in it? T this reviewer it is just a collection of formulas without specific application in the rest of the paper.

Response: Section2.1 described the configuration design model for satellite formation. For synthetic aperture radiometer based on satellite formation flight, the configuration of satellite formation affected the frequency domain sampling, which directly had the effects on the quality of the reconstructed image. Relative orbit elements were used to construct the dynamic model of satellite formation. Furthermore, the formation configuration with limited number of satellites was designed in section 4.1.

This section has been greatly modified to make it easier to understand. Please see the major revised manuscript for details.

Comment 2: Equation (7) is wrong. There is no factor 2 in the denominator \Delta T= T_s/\sqrt{B \tau_a}

Response: Thanks for pointing the mistakes, Equation (7) and (8) has been modified (Now these two equations are Equation (10) and Equation (11)).

Comment 3: Equation (14) What is T_noise? Where this noise comes from?

Response: The original meaning was to describe the brightness that was divided into true brightness and system noise e.g. receiver noise. But Equation (14) is not used in the following paper. It is deleted.

Comment 4: Equation (15) What is the "target" and what the "around". In other words, how do you define a "target"?

Response: Targets are the moving ships expected to be detected. Because of the space resolution, the targets only occupy one pixel or subpixel. These targets are point targets. In Equation (15) (The equation in the revised manuscript is Equation (12)), the Ttar is the brightness of the targets, and the “around” are the brightness in the neighborhood of the target which 11×11 pixel are selected. The purpose of defining the neighborhood is to calculate the SNR only within the neighborhood, not the SNR of the whole image. Target detection will also be carried out in the domain.

The clear explanation was added to follow the Equation (12) in the revised manuscript.

Comment 5: Line 175: the sentence "From formular (14), we know that the true image can be reconstructed perfectly when all the frequency components are sampled" is not true. You would need an instrument with infinite size. 

Response: Thanks for pointing the inaccurate description. It’s definitely right that the instrument with infinite size is needed to guarantee the measurements and image reconstruction. This part was replaced with “All the frequency components were expected to be sampled to realistic reconstruct image, from Equation (8).”

Comment 6: Line 178: "... make the image large enough that there are no sources of interest...". What are the "sources of interest". To better understand the proposed method, the authors must explain the specific application of the instrument. 

Response: In the front of section 3, the difficulties for targets detection with sparse sampling were explained. Then proposed the method. Specific application of the instrument requirements includes that Y-shaped SAIR carried by each satellite, high precision relative positioning between satellites, synchronize and inter-satellite communication, and calibration between satellites were added in Section 3 and also supplemented in Section 4.1.

Please see the major revised Section 3 for details.

Comment 7: Line 186: Why the target should move?

Response: As response to comment 4, targets were the moving ships expected to be detected. Considering the slow-varying characteristics of the observation area, and combined with the motion characteristics of the target, a target detection method based on multi-frame snapshot images was proposed.

Comment 8: Line 269: "Each satellite carries a Y-shaped SAIR, ...". How do you synchronize and syntonize the baselines across satellites?. How do you keep the antenna separation constant in those baselines?. How you calibrate the phase of all baselines?. Without all these details, this instrument, based on satellite flight formation, will not work. 

Response: Thanks for pointing these key technologies for system design.

Synthetic aperture radiometer based on satellite formation flight had several key technologies that include satellite formation design and high-precision relative position measurement, the instruments amplitude and phase calibration to keep consistency and stability, image reconstruction and targets detection on sparse sampling. Targets detection especially for point targets on sparse sampling was one of the key technologies and was the research work in our paper. The paper focused on the moving ship targets detection with sparse sampling.

In order to satisfy the interferometric requirements, the baseline measure precision, the time synchronize between satellites should be better than half of the wavelength. These all have the effected on the phase precision. Each satellite would be equipped with high stability atomic clock, GNSS receiver and also equipped with communication antenna to establish the inter-satellite link. All the daughters communicate with mother satellite. Differential positioning technology based on GNSS will be used to gain the high relative position between the satellites. All daughters will receive the calibration signal sent by mother satellite to do the amplify and phase calibration. Related engineering solution could be found in additional references.

Please see the major revised section 4.1 for details.

Comment 9: Figure 3 (a): I guess the red point is the mother and the blue points the daughter (not the other way around)

Response: Thanks for pointing the wrong description and it is be modified in Figure 3(a).

Comment 10: Line 270: Again, what are the targets?? Are they ships? So you are proposing a system to detect ships on the ocean?. You should start by this and then deduce the instrument requirements to achieve this goal.

Response: As response to the comment 4 & 7, Targets are the moving ships expected to be detected. The description was added in the “Section 3” and instrument requirements were added in “Section 4 simulation”.

Comment 11: Figure 5. Apparently the background elimination does not clearly show the positions of the targets. They are really hard to see (if seen at all) in the rightmost column.

Response: In Figure 5., ten original images as a sequency frame were showed in column (a), and two of the targets in part of the observation area were flagged with rectangles as the examples. Column (b) shows the reconstructed images by synthetic aperture radiometer based on satellite formation flight. These two targets are buried in the noise. Column (c) were the images after background elimination operation, potential targets were highlighted in the images. But they were not easy to detect. Column (d) showed the potential targets after choosing local extremum operation. Finally combined with the motion characteristics of the targets in 10 images, the targets were detected, and the trajectories were determined.

Figure 5 was modified and detected result figure was added. Please see the major revised section 4.2 for details.

Comment 12: Figure 6. What do you mean by "local extremum operation"

Response: Local extremum operation was one operation in local that remained minimum value and set the other values to zero. By local extremum operation, the noise would be suppressed, and SNR would be improved. In figure 6 (now is Figure 7 in the revised manuscript), three lines were the SNR respectively of the images reconstructed, images added background elimination operation, and images added local extremum operation.

Please see the major revised section 4.3 for details.

Comment 13: Section 6 is more a summary of the paper than the statement of conclusions.

Response: The conclusion was major revised. The innovative method for the targets detection with sparse sampling were summarized, and the SNR was improved by background elimination and local extremely operation. Spare sampling was the focus in this paper, in the future, the errors such as baselines measurement between satellites and phase errors would be added to the simulation process. Please see the revised manuscript for details.

Reviewer 2 Report

This paper proposes a target detection method for synthetic aperture radiometer based on satellite formation flight. Overall this paper is well written. Some comments are given below.

1.      In my opinion, if it is a complete detection process, the output of the filter needs to be compared with a certain threshold, as done in [r1] or [r2].

[r1]  Weijian Liu, Jun Liu, Chengpeng Hao, Yongchan Gao, and Yong-Liang Wang, “Multichannel adaptive signal detection: Basic theory and literature review,” Science China: Information Sciences, vol. 65, no. 2, Art. no. 121301, 2022.

[r2]  Weijian Liu, Jun Liu*, Tao Liu, Hui Chen, and Yong-Liang Wang, “Detector design and performance analysis for target detection in subspace interference,” IEEE Signal Processing Letters, vol. 30, pp. 618-622, 2023.

2.      If only the (4,4)the element of the matrix B in (19) is zero, while the others are all ones? If this is true, it should be clearly pointed out.

3.      The conclusion part should be written in the past tense. More importantly, the conclusion part should not only just summarize what is done, but also give useful conclusions.

Can be improved.

Author Response

Thanks for the positive and valuable comments that are helpful for revising and improving our manuscript. The changes are as following:

Comment 1: In my opinion, if it is a complete detection process, the output of the filter needs to be compared with a certain threshold, as done in [r1] or [r2].

Response: Yes, it is a complete detection process. The certain threshold was determined with the normalized reconstructive images. And we set 80 as a certain threshold in the paper. Compared to threshold of 70, the potential targets would be thousands. It was not consistent with the simulation scenario. Compared to threshold of 90, the potential targets would be dozens, it was easy to miss targets. So 80 was selected for the simulation scenario. The revised were added in Line 338 to Line 343 in Section 5. 

Comment 2: If only the (4,4) the element of the matrix B in (19) is zero, while the others are all ones? If this is true, it should be clearly pointed out.

Response: Yes, the elements in B are all ones except the center element is zero. The clear explanation was added in section 3.2 before the Equation (16) (In the previous version, it is Equation (19)).

Comment 3: The conclusion part should be written in the past tense. More importantly, the conclusion part should not only just summarize what is done, but also give useful conclusions.

Response: The conclusion was major revised. The innovative method for the targets detection with sparse sampling were summarized, and the SNR was improved by background elimination and local extremely operation. Spare sampling was the focus in this paper, in the future, the errors such as baselines measurement between satellites and phase errors would be added to the simulation process. Please see the revised manuscript for details.

Round 2

Reviewer 1 Report

The authors have responded satisfactorily to all my concerns.